# Novel Populations of *Mycobacterium smegmatis* Under Hypoxia and Starvation: Some Insights on Cell Viability and Morphological Changes

**DOI:** 10.3390/microorganisms12112280

**Published:** 2024-11-10

**Authors:** Ruben Zaragoza-Contreras, Diana A. Aguilar-Ayala, Lázaro García-Morales, Miguel A. Ares, Addy Cecilia Helguera-Repetto, Jorge Francisco Cerna-Cortés, Lizbel León-Solis, Fernando Suárez-Sánchez, Jorge A. González-Y-Merchand, Sandra Rivera-Gutiérrez

**Affiliations:** 1Departamento de Microbiología Escuela Nacional de Ciencias Biológicas, Instituto Politécnico Nacional (ENCB, IPN), Ciudad de México 11340, Mexicodi_angel_5@hotmail.com (D.A.A.-A.); aresmi26@hotmail.com (M.A.A.); jcernac@ipn.mx (J.F.C.-C.); lleons@ipn.mx (L.L.-S.); jagonzalezym@ipn.mx (J.A.G.-Y.-M.); 2Departamento de Biomedicina Molecular, Centro de Investigación y de Estudios Avanzados del Instituto Politécnico Nacional, Ciudad de México 07360, Mexico; lazaro.garcia@cinvestav.mx; 3Unidad de Investigación Médica en Enfermedades Infecciosas y Parasitarias, Centro Médico Nacional Siglo XXI, Instituto Mexicano del Seguro Social, Hospital de Pediatría, Mexico City 06720, Mexico; 4Departamento de Inmunobioquímica, Instituto Nacional de Perinatología Isidro Espinosa de los Reyes (InPer), Ciudad de México 11000, Mexico; addy.helguera@inper.gob.mx; 5Unidad de Investigación Médica en Bioquímica, Centro Médico Nacional Siglo XXI, Instituto Mexicano del Seguro Social, Hospital de Especialidades, Mexico City 06720, Mexico; fs.bioq.imss@gmail.com

**Keywords:** *Mycobacterium smegmatis*, dormancy, starvation, hypoxia, gene expression

## Abstract

The general features of the shift to a dormant state in mycobacterial species include several phenotypic changes, reduced metabolic activities, and increased resistance to host and environmental stress conditions. In this study, we aimed to provide novel insights into the viability state and morphological changes in dormant *M. smegmatis* that contribute to its long-term survival under starvation or hypoxia. To this end, we conducted assays to evaluate cell viability, morphological changes and gene expression. During starvation, *M. smegmatis* exhibited a reduction in cell length, the presence of viable but non-culturable (VBNC) cells and the formation of anucleated small cells, potentially due to a phenomenon known as reductive cell division. Under hypoxia, a novel population of pleomorphic mycobacteria with a rough surface before the cells reached nonreplicating persistence 1 (NRP1) was identified. This population exhibited VBNC-like behaviour, with a loss of cell wall rigidity and the presence of lipid-body-like structures. Based on *dosR* and *hspX* expression, we suggest that *M. smegmatis* encounters reductive stress conditions during starvation, while lipid storage may induce oxidative stress during hypoxia. These insights into the heterogeneous populations presented here could offer valuable opportunities for developing new therapeutic strategies to control dormant mycobacterial populations.

## 1. Introduction

There is increasing interest in studying dormant and persistent mycobacteria due to their role in treatment failures and disease relapses in infected patients. The World Health Organization estimates that a quarter of the world’s population has been infected with *Mycobacterium tuberculosis* (*M. tuberculosis)* [1]; its persistence and lethality are largely due to its ability to develop a latent infection where this bacterium remains dormant for years [2]. In general, dormant mycobacteria are defined as a population of mycobacteria that have very low metabolic activity, remain viable in the host for prolonged periods, show a drug-tolerance or resistance profile at any time and, in some cases, include viable but nonculturable (VBNC) microorganisms [3]. Dormant mycobacteria have been studied in several in vitro models involving an acidic environment, starvation and the stationary phase of growth, oxidative stress, depletion of potassium, the prolonged presence of certain drugs, the presence of lipids, and the gradual depletion of oxygen [4,5,6,7,8,9,10,11,12,13].

The transition to dormancy is controlled at both the transcriptional and translational levels through mechanisms such as the activation of efflux pumps, the glyoxylate shunt, and toxin-antitoxin modules. Additionally, it involves the maintenance of intracellular redox balance, deposition of triacylglycerols in the cytoplasm (forming lipid bodies), reductive cell division, decreased membrane fluidity, and bacterial growth arrest [14,15,16,17]. All these alterations result in significant changes in cell morphology, such as those described in *M. tuberculosis*: the thinning of the cell wall, which leads to the loss of mycobacterial acid fastness, and the subsequent formation of L-form variants or changes in the thickness of the cell wall, which produce ovoid cells, are specifically shown in mycobacteria surviving under starvation and acidic conditions. However, these changes are not always detected in all in vitro models of dormancy [18,19].

Models of dormancy that analyse entire populations combined with single-cell technologies have been crucial for understanding the heterogeneity of dormant populations [20,21,22]. Continued research using these technologies is essential for identifying effective targets to control latent infections. This work aimed to describe the physiological and morphological changes associated with bacillus resilience during adaptation to dormancy using two in vitro models: starvation and hypoxic conditions. We employed flow cytometry (FCM), transmission electron microscopy (TEM), scanning electron microscopy (SEM), and gene expression analysis to study *Mycobacterium smegmatis* (*M. smegmatis*), a model organism commonly used for investigating the general biological properties of mycobacteria [16].

## 2. Materials and Methods

### 2.1. Bacterial Strain and Growth Conditions

*Mycobacterium smegmatis* mc2 155 was cultured in Difco^TM^ Dubos Broth (casein enzymic hydrolysate and L-aspargine, inorganic salts, and polysorbate 80 (USA), supplemented with 10% ADC supplement: 0.2% (*w*/*v*) dextrose, 0.2% (*v*/*v*) glycerol, 0.5% bovine serum albumin, catalase (4 mg/L) and 15 mM sodium chloride (BD, USA). Cells were grown at 37 °C and shaken at 200 rpm with a stainless-steel spring as previously described [23] until the exponential growth phase was reached at 20 h of incubation with an Optical Density at 640 nm (OD_640_) equal to 1.0 ± 0.05.

### 2.2. Starvation Model of Dormancy

To perform the starvation model of dormancy, 100 mL of *M. smegmatis* cultures in the exponential phase were centrifuged and resuspended in 100 mL sterile phosphate-buffered saline (PBS) with 0.05% Tween 80 contained in a 250 mL conical flask with a stainless-steel spring. The flask was incubated at 37 °C and shaken at 200 rpm, as described previously [6]. Samples were taken at 0 h, 4 h, 12 h, 24 h, and 120 h of incubation under starvation conditions. To perform all experiments from these time lapses, three independent batches of cells (biological replicates *n* = 3) were carried out.

### 2.3. Hypoxic Model of Dormancy

To perform the in vitro model of dormancy based on hypoxia, 100 mL cultures in the exponential phase were centrifuged and resuspended in 200 mL of Dubos medium supplemented with ADC and then subjected to the Wayne and Hayes model until they reached non-replicative persistence 1 (hypoxic conditions, in which the oxygen concentration in the culture is approximately 1%) and non-replicative persistence 2 (anaerobiosis, in which the oxygen concentration is less than 0.06%) phases of dormancy, as described previously [4,24,25,26]. Parallel cultures were set up with methylene blue (MB) (1.5 µg/mL) as an indicator of oxygen depletion. The fading of methylene blue indicated a low respiratory rate of mycobacteria and thus an association with low metabolism and the hypothetical existence of the non-replicative persistence stages (NRP) described by Wayne and Hayes [4]. The NRP1 phase (initial fading of MB) was determined at 60 h after the exponential phase, and the NRP2 phase (complete discolouration of MB) was determined after 110 h. Samples from the hypoxic model (without MB) were taken at 12 h, 36 h, 72 h, and 120 h of incubation. To perform all experiments from these times, three independent batches of cells (biological replicates n = 3) were analysed.

### 2.4. Growth Measurements and Bacterial Viability Determinations

*M. smegmatis* growth was monitored using the OD_640_, by counting the colony-forming units per millilitre (CFU/mL) and by flow cytometry (FCM) as described previously [23]. In brief, 10-fold serial dilutions of cultures (10^−1^ to 10^−6^) were performed in tubes containing 0.05% Tween 80 and 300 µL glass beads and were plated on Middlebrook 7H11 agar (Becton Dickinson, Franklin Lakes, NJ, USA). The plates were incubated at 37 °C for five days before quantification of colonies (CFU/mL).

### 2.5. Flow Cytometry Analysis

Regarding FCM, the total number of bacteria and cell viability were determined using the Live/Dead BacLight Viability and Counting Kit (Molecular Probes Inc., Eugene, OR, USA), which contains a microsphere standard of 0.6 µm and the fluorochromes SYTO 9 and propidium iodide (PI) as markers for viable and non-viable (membrane-compromised) cells, respectively. According to the supplier’s instructions, the cell concentration of the samples was adjusted between 10^6^–10^8^ cells in a final volume of 1 mL; to this suspension, 6 µL of an equimolar solution of components A and B were added [component A (9.86 µM SYTO 9 final concentration) and component B (59 µM Propidium Iodide final concentration)] as well as 10 µL of the glass bead standard (6.0 µm), and the cell population was allowed to incubate for 15 min in the dark with shaking. Kohn-Harris medium [27] without succinate was used as a diluent to adjust the cell concentration of the samples. As mentioned, the kit includes a 0.6 µm microsphere standard, which serves as a reference for calibrating the flow cytometer to accurately quantify the total number of bacterial cells. The microsphere count helps determine the concentration of bacterial cells in the sample by comparing the number of bacterial events to the known concentration of microspheres. The total bacterial cell count is calculated using a formula provided by the kit’s manufacturer, which incorporates the detected fluorescent signals (events) and the microsphere standard.

The formula for this determination is as follows:$$\begin{array}{c} Concentration\;of\;bacteria\left(\frac{cells}{mL}\right)\\ =\left(\begin{array}{c} Number\;of\;bacterial\;events\frac{count}{Number}\\ of\;microsphere\;events\;counted \end{array}\;\right)\\ \times \left(Concentration\;of\;microspheres\frac{\frac{cells}{mL}}{Volume}of\;sample\;analysed\left(mL\right)\right) \end{array}$$

Concentration of bacteria (cells/mL) = (Number of bacterial events counted/Number of microsphere events counted) × (Concentration of microspheres (cells/mL)/Volume of sample analysed (mL)).

Samples were analysed using a BD FACSCalibur^TM^ flow cytometer (Becton Dickinson, USA) at a flow rate of 12 µL/min using the FACSFlow^TM^ Sheath fluid (Becton Dickinson, USA). In addition to forward scatter (FSC) and side scatter (SSC), SYTO 9 fluorescence was captured by excitation at 488 ± 15 nm (green), and PI fluorescence was captured by excitation at 585 ± 21 nm (red). Because of the small size, all signals were detected by using logarithmic amplification of intensities, and on each occasion, 100,000 events were analysed. Data were evaluated with the CellQuest programme (version 3.1f, Becton Dickinson). In these experiments, the data represent three independent batches of cells (biological replicates) that were each examined in triplicate (assay replication, n = 9).

### 2.6. Transmission Electron Microscopy

Cells were harvested by centrifugation at 13,000× *g*, fixed in 2.5% glutaraldehyde/0.1 M sodium phosphate (*v*/*v*) for 1 h at room temperature and washed twice in a solution containing 1.1% Na_2_HPO_4_, 0.25% NaH_2_PO_4_/H_2_O, 10% sucrose and 0.1% CaCl_2_, at a final pH of 6.7. Cells were recovered and resuspended in 1% OsO_4_ (*w*/*v*) for 1 h and washed in PBS. Samples were dehydrated by treatment in a series of ethanol solutions (30%, 40%, 50%, 60%, 70%, 80%, and 90%; each treatment was performed twice for 10 min). A final dehydration step was performed twice with absolute ethanol for 10 min and once more for 30 min. Samples were embedded in an EPON LR White 812 resin (EMS, Hatfield, PE, USA). This resin was polymerised at 60 °C for 24 h. Resin blocks were cut in a LEICA Ultracut UCT ultramicrotome (Leica, Wetzlar, Germany) to a 70 nm thickness, placed on copper grids (200 mesh, EMS) and stained with 30% uranyl acetate (*w*/*v*)/70% ethanol (*v*/*v*) and counterstained with Reynold’s lead citrate buffer. Samples were examined using a Jeol JEM-1010 Transmission Electron Microscope (TEM) (USA) with an acceleration voltage of 60 kV. Magnifications were at 10,000×, 120,000×, and 150,000×. Five random fields were considered from each sample in three independent experiments.

### 2.7. Scanning Electron Microscopy

Cells were harvested and processed as for TEM; after dehydration with ethanol solution, samples were dried in hexamethyldisilazane (HMDS; EMS, USA). Samples were gold-coated using a Desk II Denton Vacuum at 100 mTorr and 10–20 mA for 400 s and examined using a Jeol JSM-S800 LV Scanning Electron Microscope (SEM) (USA) with an acceleration voltage of 15 kV. Magnification was from 10,000× to 40,000×. Five random fields were considered from each sample in three independent experiments.

### 2.8. RNA Isolation and cDNA Synthesis

Total RNA was calculated from each culture condition (exponential phase, starvation, or hypoxic conditions). Basically, cultures were harvested by centrifugation and pellets were resuspended in guanidinium chloride buffer in a proportion of 1 mL of buffer/100 mL of culture. Cells were lysed mechanically in a FastPrep (Thermo Scientific, Waltham, MA, USA) with glass beads (Sigma-Aldrich, St. Louis, MO, USA) by performing four lysis cycles of 15 s each at high speed (6.5 m/s). Nucleic acids were purified with phenol-chloroform-isoamyl alcohol (25:24:1), and RNA was differentially precipitated with 0.4 volume of absolute ethanol. Finally, RNA was purified three times with Trizol™ reagent (Invitrogen, Waltham, MA, USA) to eliminate DNA remnants. RNA integrity was analysed with a bioanalyser (Agilent Technologies, Singapore) and quantified by spectrophotometry with the NanoDrop (Thermo Scientific) [28]. cDNA was prepared using 1 µg of RNA, random hexamers (0.5 mg/mL), and the SuperScript First Strand cDNA Synthesis Kit (Invitrogen, USA), following the manufacturer’s instructions.

### 2.9. Quantitative Real-Time PCR

Four genes involved in cell division (*dnaA* and *ftsZ*) and dormancy/stress (*hspX* and *dosR*) were quantified by qRT-PCR using gene-specific primers previously described [26] and the FastStart DNA Master SybrGreen I Kit (life Science, Roche) in a LightCycler 2.0 thermocycler (Life Science, Roche) was used. Three biological and nine technical replicates were used to ensure reproducibility; 16S rRNA expression was used for normalisation, with satisfactory results [26]. Primer sequences were: 16SF: 5′ATGACGGCCTTCGGGTTGTAA-3′ and 16SR: 5′-CGGCTGCTGGCACGTAGTTG-3′, dnaAF 5’-CGTTCAAGCGCAGCTACCG-3´, dnaAR 5´-AGATGGCCGAGCGCCGTGAGGTGTT-3´; ftsZF 5´-GCAGTGCCTGGGCATCGGTGTT-3´, ftsZR 5´-TCGCGGTCAATCAAGGTGGTT-3´; hspXF 5´-GGTGAATCCCTTGAACCAGTC-3´, hspXR 5´-AGCACGCTGATGAAGACGC-3´; dosRF 5´-GCTGGCGAAACTCGGGATG-3´, dosRR 5´-TTGCTCGTCGTGGTGGC-3´.

To quantitatively analyse gene expression differences between the exponential phase and stress conditions (starvation or hypoxia), logarithmic graphics of the expression of each gene (reported as copies of the gene/µg RNA) were generated. Finally, to compare the expression obtained at each time point under starvation or hypoxic conditions, the expression relative to that in the exponential phase was calculated; this ratio shows how many times each gene is overexpressed or downregulated in each condition relative to the exponential phase of growth.

### 2.10. Statistical Analyses

Cell length determined from TEM micrographs was obtained by analysing at least 50 cells. Other morphological changes determined by SEM micrographs were considered by viewing at least five random microscopic fields. Means and standard deviations were calculated for all data. Student’s *t*-test was used to statistically determine significant differences with a 99% confidence interval between cell sizes obtained by TEM. Significant differences between populations observed in flow cytometry histograms were calculated using the Kolmogorov–Smirnov test; we considered a value statistically significant only when (α) = 0.01.

One-way ANOVA with Tukey’s multiple comparison procedure was used to determine the significant differences between different growing conditions. GraphPad Prism V 8.0.2 was used to calculate *p*, with *p* < 0.05 considered statistically significant.

## 3. Results

### 3.1. Growth Measurements and Cell Viability of M. smegmatis

Our study reveals that *M. smegmatis* mc2 155 survived and persisted during 120 h of starvation or hypoxic conditions (Figure 1); during this time, several viability and morphological changes occurred: a membrane compromise of the *M. smegmatis* was identified by FCM experiments (Figure 2 and Figure 3), cell length reduction and the appearance of pleomorphic and rough cells were also identified (Table 1 and Table 2). During the first 24 h of starvation, no correlation of OD with CFU/mL or FCM methods was observed: while OD increased steadily, the amount of CFU/mL and viable bacteria (v-bact) diminished after 4 h and 24 h of incubation, respectively (Figure 1A). From 24 h onward, all measurements remained relatively constant (Figure 1A).

The cell viability results deserve special attention, since the CFU/mL and v-bact determined by FCM were similar at the beginning of the experiment, with around 5.3 × 10^7^ CFU/mL and 5.0 × 10^7^ v-bact/mL, respectively. The levels of v-bact/mL were then higher than those of CFU/mL between 4 h and 12 h of starvation, which indicated that during that period, some cells were viable but unable to form colonies on agar plates, indicating the presence of putative viable but not culturable (VBNC) cells at the beginning of the starvation conditions. Figure 1A also shows a sharp decline in CFU/mL during the first 4 h of starvation; following this initial drop, CFU/mL values began to rise. By contrast, the v-bact/mL detected by FCM decreased and reached its lowest level at 24 h.

However, when *M. smegmatis* cells were subjected to hypoxia, the initial methylene-blue fading (in parallel cultures) was identified at 60 h of incubation (NRP1 phase), and total discolouration was observed at 110 h (NRP2 phase). This model highlighted a direct correlation between OD measurements and the number of total bacteria (detected by FCM) during the 120 h of cell adaptation to hypoxia. Nevertheless, this was not the case when the OD and CFU/mL data were compared (Figure 1B).

The amounts of v-bact/mL measured by FCM plotted in Figure 1B show a clear decrease at 36h during hypoxia; however, a new population of bacteria was detected (which was absent in starvation). This particular population exhibited the same intensity of fluorescence coming from the two fluorochromes used in FCM to detect both v-bact and total bacteria (t-bact/mL). These singular results will be further described as a part of the FMC analysis in the following paragraphs, as well as the suggested presence of membrane-compromised cells (MCC) and VBNC mycobacteria without membrane compromise proposed in starvation or hypoxic conditions, respectively.

### 3.2. Multiparametric Analysis of Mycobacterial Growth Obtained by FCM

Our FCM results provided insights into various features of *M. smegmatis* cells grown under starvation and hypoxia conditions. These features included cell complexity, size, envelope permeability, and viability. The Live/Dead BacLight Viability kit, which uses the fluorochromes SYTO 9 and PI, was used to stain the DNA of viable and MCC, respectively.

At the beginning of the starvation condition, two populations of cells were detected: viable cells and dead cells (Figure 2A, 0 h and 4 h). The small portion of dead cells observed at the beginning of the experiment increased over the study period. However, at 12 h, there was a minimum population of dead cells, which notably increased by the next time point (24 h) (Appendix A). At the end of the experiment (120 h), it was difficult to distinguish between dead and live cells because both populations acquire the same behaviour based on staining with these fluorochromes. We hypothesise that some bacteria fail to sustain growth that occurs during the first 12 h of starvation; after that, the dead population observed at 24 h could correspond to membrane-compromised VBNC mycobacteria, which is accentuated until 120 h (Figure 2A, 120 h).

Figure 2B highlights the permeability profiles of *M. smegmatis* under starvation by charting the fluorescence of SYTO 9 (FL1-H) over time. While there was a maximum peak in the fluorescent channel (FL1-H) of 300 at time 0, at 120 h, the maximum peak was detected at 15 on the FL1-H axis. Considering these findings in the reduction in the permeability of SYTO 9 as well as the dot plots (Figure 2A), we propound the idea that there is an increase in membrane compromise vs. time under starvation conditions, meaning that at 120 h under starvation, *M. smegmatis* cell permeability increases. Both fluorophores internalised after cell-envelope compromise, and PI caused a decrease in SYTO 9 emission by replacing it in the nuclei acid interaction, resulting in a possible population of VBNC mycobacteria (Figure 2A 120 h and Figure 2B).

Figure 2C,D also reveal that cell size and complexity decreased over time under starvation. However, the Kolmogorov–Smirnov statistical test proved that while the change in cell size was significant (α = 0.01), the complexity/granularity was not (α ≠ 0.01). These findings suggested that cells were dividing at 0 h and 4 h (in starvation), as judged by the broad range of cell size distribution.

By contrast, the viability analysis of *M. smegmatis* during its adaptation to the gradual depletion of oxygen (hypoxia dormancy model) performed by FCM displayed three populations according to their staining properties: the presence of viable cells (v-bact), dead cells (d-bact) and cells in an intermediate stage between live and dead bacteria (SYTO 9^high^PI^high^-bact). Figure 3A shows v-bact stained with SYTO 9 and a small population of d-bact stained with PI throughout the experiment. At 36 h of hypoxia, before the NRP1 phase, an additional population of ‘SYTO 9^high^PI^high^-bact’, which presented the same fluorescence intensity with both fluorochromes (PI and SYTO 9), was detected by FCM and remained until 120 h of incubation (Figure 3A). These bacteria had the same staining properties as the population at 120 h of starvation according to its distribution on the dot plots (SYTO 9/PI, Figure 2A), but they were larger than those in starvation. Those three populations were quantified, and the comparison of the viable bacterial (v-bact/mL) counts performed by FCM with the CFU/mL in agar plates is displayed in Figure 3B.

The cell numbers measured by FCM were similar to the CFU/mL counts only during the first 12 h. Subsequently, at 36 h, the number of v-bact declined by 87% (from 3 × 10^7^ to 3.9 × 10^6^ v-bact/mL), and at the same time, the population of SYTO 9^high^PI^high^ -bact began to appear at a level close to the CFU/mL (about 4 × 10^7^ SYTO 9^high^PI^high^/mL and 6.5 × 10^7^ CFU/mL). The result is even closer if we add the v-bact and the SYTO 9^high^PI^high^-bact (Figure 3B bar graph), suggesting that the SYTO 9^high^PI^high^-bact and v-bact could form colonies in solid media at that moment.

Once the dissolved oxygen in the media dropped below 1% (as indicated by the initial fading of MB), the population of SYTO 9^high^PI^high^-bact was 2.5 times higher than either that of v-bact or the CFU/mL from 72 to 120 h. This result suggests that SYTO 9^high^PI^high^-bact was likely viable but not culturable (VBNC) after 36 h under hypoxic conditions. Furthermore, the size and granularity of SYTO 9^high^PI^high^-bact and v-bact were not statistically significant (α ≠ 0.01) as shown in the data (Figure 3C,D). This indicates that this SYTO 9^high^PI^high^ population had a similar cell-envelope compromise to the population at 120 h under starvation, exhibiting increased cell permeability according to FCM data albeit with a larger size. Given their increased prominence over time, this SYTO 9^high^PI^high^ population may correspond to cells that are well adapted to hypoxic conditions. As shown in Figure 3B, we quantified viable bacteria and those adapted to hypoxia using CFU/mL measurements. The data indicate that at 36 h in the Wayne model, the bacteria are capable of multiplying similarly to traditionally viable bacteria. This suggests that adaptation to hypoxia leads to a population of viable bacteria exhibiting different permeability characteristics, as indicated by the staining results. At this 36 h mark under hypoxic conditions, the adapted population, identified by SYTO 9^high^ and PI^high^ staining, is larger than the classical viable and dead bacterial populations.

Even if there are VBNC bacteria in both dormancy models (starvation and hypoxia), they have different physiological and morphological characteristics; for these reasons, we analysed the changes in the ultrastructure of *M. smegmatis* by TEM and SEM experiments as well as the expression of some genes related to dormancy.

In the hypoxia model, the most significant changes become evident after 36 h; due to the hermetically sealed environment, there is no air exchange in the medium, causing bacterial metabolism to deplete oxygen from the outset. As a result, adaptation to the stress conditions occurs gradually. Therefore, we chose 36 h as the clear point at which distinct staining populations can be observed.

### 3.3. Gene Expression

We evaluated the gene expression levels of the following genes: *dnaA*, which encodes DnaA, a protein involved in bacterial chromosome replication; *ftsZ*, which encodes FtsZ, a protein involved in septum formation; *dosR*, which encodes the two-component transcriptional regulatory protein DevR; and *hspX*, which encodes the heat shock protein HspX (an alpha-crystallin homolog) [29,30].

During starvation, we observed a downregulation of *dnaA* and *ftsZ* (Figure 4), suggesting the presence of anucleated small cells resulting from the reductive cell division process. Unlike in eukaryotes, the bacterial cell division cycle does not consist of discrete stages; instead, DNA replication and the assembly of the division machinery overlap, with DnaA and FtsZ being essential proteins in these processes [29,30]. This finding may partially explain the differences observed in the CFU/mL and FCM results (Figure 1). In contrast, *dosR* was upregulated only at 4 h, after which its expression abruptly decreased, while *hspX* was upregulated solely at 120 h (Figure 4).

Under hypoxic conditions, *dnaA* was consistently downregulated, whereas *ftsZ* was upregulated only at 4 h, followed by a decrease in expression. In this context, *dosR* was upregulated at all time points, while *hspX* showed upregulation until 36 h (Figure 4).

It was found that the presence of anucleated cells (cells without nucleoid) in a starvation environment is linked to the downregulation of the *ftsZ* gene, leading to a decreas e in the FtsZ protein and an increase in small cells due to inhibited cell division. Similarly, the presence of anucleated cells is also associated with the downregulation of the *dnaA* gene, resulting in a reduced amount of DnaA and cells lacking nucleoids. These findings provide a deeper understanding of the intricate role of FtsZ and DnaA in bacterial growth and division and their implications in survival strategies under adverse conditions.

Likewise, under hypoxic conditions at 12 h, it was observed that the *dos*R and *hsp*X genes were over-expressed during hypoxia; as time passed, the expression of these genes decreased (Figure 4C,D), and it was observed that *dos*R and *hsp*X genes were expressed 1 × 106 fold. These genes are expressed when the mycobacteria is under a state of hypoxia: the *dos*R hypoxia regulator ensures the survival of the mycobacteria during hypoxia-induced in vitro latency; while the *hsp*X gene is induced in low oxygen tension, the HspX protein also helps the mycobacteria to survive during non-replicative persistence phases.

### 3.4. Morphological Changes in Dormant M. smegmatis Examined by Electron Microscopy

The analysis of SEM and TEM images of these bacteria at different times (in each stress condition) and their comparison with micrographs of mycobacteria in the exponential phase of growth opens up new avenues for understanding the adaptive mechanisms of these bacteria.

During starvation, the main change detected by TEM was a significant decrease in bacterial length, which declined sharply (size reduction of 25%) within the first four hours of starvation (Table 1). At 24 h, a few rods began to exhibit a rough appearance, while at 120 h the cell length was reduced by 63%, and all cells had a rough surface (Table 1 and Figure 5).

At 12 and 24 h of starvation, we reported a few electron-dense cells without nuclei (Figure 5), confirming the presence of anucleated small cells that increased the OD values during starvation, as well as the presence of little putative detritus at 24 h, which increased in quantity at 120 h of starvation (Figure 5A). These cell remnants could have resulted from cellular apoptosis when *M. smegmatis* sensed nutrient limitation. These changes in ultrastructure are better displayed in Figure 5B, where micrographs show an increased electron-dense bacterial cytoplasmic content over time, and the nucleoid became compacted and surrounded by an electron-transparent zone at 120 h.

Using SEM, we show the presence of bacilli with rough surfaces and many septa from 24 h to 120 h (Figure 6A,B), which might be MCC. As shown in Figure 1A, all measurements of cell growth remained constant from 24 h to 120 h, probably due to the balance between dead and recently divided cells judging by the presence of septa observed in Figure 6 and by the presence of cell aggregates and cell debris at 120 h (Figure 5A, 120 h).

According to the mRNA expression data, *M. smegmatis* maintained low expression of genes promoting cell division. However, at 120 h, the expression of *dnaA*, a key gene in bacterial replication, became more active. This suggests that the bacilli detected the availability of some nutrients from the dead cells, triggering a restart of its active growth. The downregulation of *ftsZ* further explains the presence of many septa (Figure 4).

In the model of hypoxia, regular rods with a smooth appearance to their surface were the predominant cell shape (75%) observed at 12 h of hypoxia (Table 2). After this, two changes occurred: smooth-surfaced bacilli were transformed into pleomorphic and rough-surfaced bacteria. These findings are novel and add to our understanding of bacterial responses to hypoxia. The pleomorphic transformation was first detected at 12 h of incubation, and rough-surfaced bacteria were first seen at 36 h under hypoxic conditions. Regular rods represented only 16–18% of the total population from 36 h to 120 h, and the cell length of these did not change significantly over time (Table 2). By contrast, pleomorphic cells showed considerable variation in shape and size. These pleomorphic cells seem to be the precursors of rough-surface cells (Figure 7, 72 h and 120 h).

We suggest that the pleomorphic cells observed in Figure 7 correspond to SYTO 9^high^PI^high^-bact, since the counting of both kinds of cells (SYTO 9^high^PI^high^-bact and pleomorphic cells) demonstrated that they were the most significant population at 36 h of hypoxia (Figure 3B and Table 2). Thus, these SYTO 9^high^PI^high^-bact detected by FCM were uniquely present in the hypoxic model and were neither present in the aerobic growth nor in the starvation condition. These SYTO 9^high^PI^high^-bac probably give rise to rough cells, because we found a relatively constant population of SYTO 9^high^PI^high^-bact from 36 h to 120 h (Figure 3B), meaning that rough morphology increases as a response to oxygen depletion.

During this period (36 to 120 h), we observed the presence of large, lipid body-like structures (LB) (Figure 8), which, in turn, could contribute to the transformation of regular rods into pleomorphic cells.

TEM micrographs showed that these LB structures were detected mostly in pleomorphic cells and increased in number and size as the level of oxygen diminished. Some cells were detected with a less electron-dense cytoplasm (Figure 8A,B). A closer view of these cells (Figure 8B) revealed important changes in ultrastructure that involved the supercoiling of DNA, like compacted and highly electron-dense DNA, surrounded by a transparent-electron zone, which was consistently found together with LB structures after 36 h. Additionally, a population of cells with a star- or network-type nucleoid, mainly at 120 h of incubation, was found (Figure 8B). Other observed changes were the loss of rigidity of the cell wall at 72 h and 120 h and the presence of cells with membrane invaginations that gave rise to vesicle-like structures (Figure 8C).

## 4. Discussion

Many researchers have proposed that dormant mycobacterial populations are heterogeneous [21,31,32,33], and our study supports this. We found that starvation, as well as hypoxia, induced the formation of cell populations that contribute tolerance to these stress conditions.

At the beginning of the starvation period (24 h), the decreased CFU values and the increased OD values reported here correlated with similar results previously observed in stationary-phase cultures of *M. smegmatis* [34]. Although the reason for this change is not yet fully understood, we can hypothesise that it might be related to an enhancement of cellular mass. A specific phenomenon known as ‘reductive cell division’ has been described previously [35], in which a putative bacterial division event produces two types of cells: nucleated bacteria and ‘anucleated’ ones. The anucleated cells self-degrade but still contribute to the cell density detected by the OD of the medium, explaining the lower CFU values.

Additionally, the sharp decline in CFU values may be related to the initiation of ‘programmed cell death (PCD)’, a form of cellular apoptosis observed in bacteria within high-density cultures when nutrients are scarce [2]. Since not all bacteria undergo this form of apoptosis, the surviving bacilli, sensing the low bacterial density, can exit from this ‘PCD’ state and resume growth, albeit at a reduced growth rate (Figure 1A, 24 h to 120 h). As we can observe after 4 h, the increase in CFU/mL values might indicate the likely presence of viable membrane-compromised cells (MCC) that were marked in FCM experiments as dead cells but can re-grow and form colonies when they are plated on a rich medium, like L-forms previously described in *M. bovis* BCG and *M. tuberculosis* [18,36]; however, isolation of this population by FACS to perform a better transcriptomic characterisation of these cells is needed as well as a further analysis with the geometric mean of all events detected, as in all times more than one peak was observed.

Some of the results presented here align with the previously described starvation-induced transcriptional differentiation programme of *M. smegmatis*, using a starvation model with PBS (zero-nutrient) and PBS-Tween 80-oleic acid (which provides traces of a carbon source), Wu and collaborators (2016) identified Large Resting Cells (LARCs) under zero-nutrient conditions, while in the presence of trace carbon sources, they observed Small Resting Cells (SMRCs) [37]. The morphological characteristics outlined here, including the presence of septated cells and the upregulation of the *ftsZ* gene during the first 4 h of starvation, correspond with the data reported for LARCs. After 24 h, the observed reduction in cell length is consistent with the characteristics of SMRCs. Additionally, our work provides insights into the ultrastructure, revealing changes in chromosome compaction that may be linked to the increased expression of the *Hlp* gene expressed in both types of resting cells (as Wu and collaborators proposed); however, this hypothesis requires further investigations.

In the other model, under hypoxic conditions, the lack of correlation between OD measurements and CFU/mL values, despite the number of total bacteria detected by FCM, leads us to propose two possible scenarios: (1) *M. smegmatis* may alter its cell wall composition and increase neutral-lipid storage, which thickens the outer layer, as previously reported for *M. tuberculosis* [38]; (2) There could be a population of VBNC mycobacteria that, unlike in starvation conditions, do not exhibit membrane compromise. This would explain the good correlation between OD and FCM (total-bac) values. Additionally, the maintenance of a homogeneous mycobacterial population of non-complex, small rods during starvation aligns with previous reports for *M. smegmatis* and other bacteria. These studies suggest that the size of microorganisms is influenced by nutrient availability [34,39].

We are aware that it is crucial to interpret the results with caution when using SYTO 9 and PI staining, as viable cells might be incorrectly identified as dead when their membranes are compromised during cell division, cell wall synthesis, or stress [40]. Furthermore, it has been reported that mycobacterial species can incorporate a significant amount of PI, up to 45%, through porins when grown under aerobic conditions [41]. This finding, along with the demonstration by Aguilar-Ayala and co-workers (2017) of the overexpression of the porin main gene (*ompA*) of *M. tuberculosis* during hypoxia [15], opens intriguing possibilities. Another important aspect to consider when using SYTO 9 staining is that the observed results can be attributed to cytoplasmic leakage of the stain or increased activity of efflux pumps that expel SYTO 9 from the cells [42]. It suggests a potential hypothesis that the these SYTO 9^high^PI^high^-bact probably give rise to rough cells, because we found a relatively constant population of SYTO 9^high^PI^high^ -bact from 36 h to 120 h. The -bact population detected under hypoxia could be a dormant mycobacterial population with many porins, facilitating the entry of PI along with SYTO 9. These findings could have far-reaching implications for our understanding of mycobacterial behaviour. SYTO 9^high^PI^high^-bact populations have been described as only present in Gram-negative bacteria, since small counts of SYTO 9^high^PI^high^-bact of *S. enterica* serovar Typhimurium and *S. flexneri* have been found when using the LIVE/DEAD BacLight kit [43,44]. According to those studies, they might have a mildly compromised membrane, and when conditions become favourable, they might grow again [45] as the MCC detected under starvation, promoting cell survival and the recovery of viability and persistence. In this way, *M. smegmatis* could be employing a similar strategy to that observed during an active infection in nutrient-starved, *M. tuberculosis*-infected mice, where this mycobacterium was genetically identical to an exponentially growing *M. tuberculosis* but with a breakdown of protective antimycobacterial immunity [46]. Therefore, more studies focusing on this *M. smegmatis* population with a compromised cell membrane are necessary to correlate its importance during cell growth renewal and persistence, making it possible for the persistence of other heterogeneous populations through cooperative behaviours, as has been previously observed in mycobacteria [47].

Furthermore, during hypoxia, a novel population of pleomorphic mycobacteria with a rough surface and lipid bodies was identified after 36 h, increasing in size over time. As we also noticed substantive changes in the hypoxic model just before and after the initial fading of methylene-blue (NRP1), we believe that *M. smegmatis* cells sense the redox state of the medium; as such, a reduced environment might be the primary signal that triggers the long-term survival of mycobacteria during stress conditions. The same effect regarding gene expression of *dosR* and *hspX* has been reported to occur when *M. tuberculosis* is grown under hypoxia [48]. Nevertheless, in this work, these changes occurred before NRP1, whereas those in ultrastructure were reached after NRP1 and well expressed until NRP2.

Viable but not culturable mycobacteria were detected in both models with compromised membranes, in which, unlike in dormant *M. tuberculosis*, we did not find thickening of the cell envelope. However, the presence of LB here corroborates a similar strategy of *M. smegmatis* to *M. tuberculosis* to persist in the same way gene expression distinguishes the two stress conditions in which this mycobacterium survives. Previously, it was reported in *M. tuberculosis* that the downregulation of *dosR* and the upregulation of *hspX* were associated with reductive stress conditions. By contrast, the upregulation of *dosR* and *hspX* accompanied oxidative stress conditions [11,25]. During starvation, *M. smegmatis* could face reductive conditions probably generated after the ‘reductive cell division’ process, in which lipid metabolism could be enhanced (as a sole carbon source) as the adapted bacteria metabolised lipids from the dead cells. Under hypoxic conditions, *M. smegmatis* might behave like *M. tuberculosis* when cultured with dextrose, where cells store lipids rather than using them as a carbon source. Previous studies showed that the presence of lipid cytoplasmic inclusions is highly related to dormant mycobacteria and especially well described in dormant *M. tuberculosis* [11,14].

In conclusion, our study highlights the significance of specific morphological changes in mycobacterial populations under stress conditions, providing valuable insights for identifying novel therapeutic targets. Both starvation and hypoxia induce the formation of distinct cell populations crucial for survival in these adverse environments. Particularly under hypoxia, the observed changes occurring before *M. smegmatis* reaches the NRP1 phase should be considered when developing therapeutic strategies. The immune system is likely to encounter these early-stage phenotypic adaptations during latent infections, making them critical targets for controlling both active and dormant mycobacterial populations.

## Figures and Tables

**Figure 1 microorganisms-12-02280-f001:**
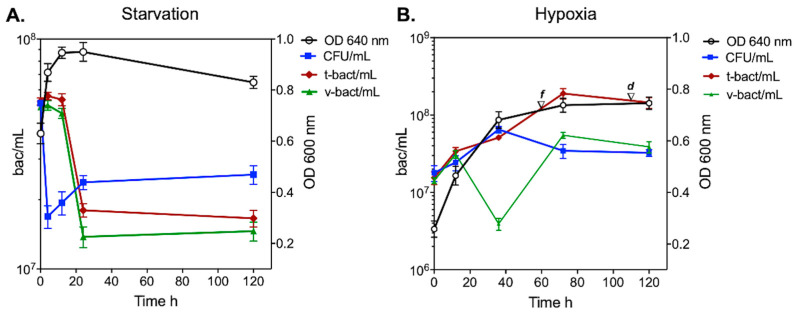
Correlation between growth measurement and cell viability methods of *M. smegmatis*. (**A**) Growth and cell viability measurements during starvation. (**B**) Growth and cell viability measurements during hypoxia. White circles represent the optical density (OD), blue squares represent the number of CFU/mL, red diamonds represent total bacteria/mL (t-bact/mL) detected by FCM, and green triangles represent viable bacteria/mL (v-bact/mL) detected by FCM. The time when NRP1 (initial fading of the methylene blue, ▽f) and NRP2 phases (total discolouration of the methylene blue occurred, ▽d) began are indicated. Values are displayed on a logarithmic scale and mean ± SD are plotted. All experiments were carried out in triplicate.

**Figure 2 microorganisms-12-02280-f002:**
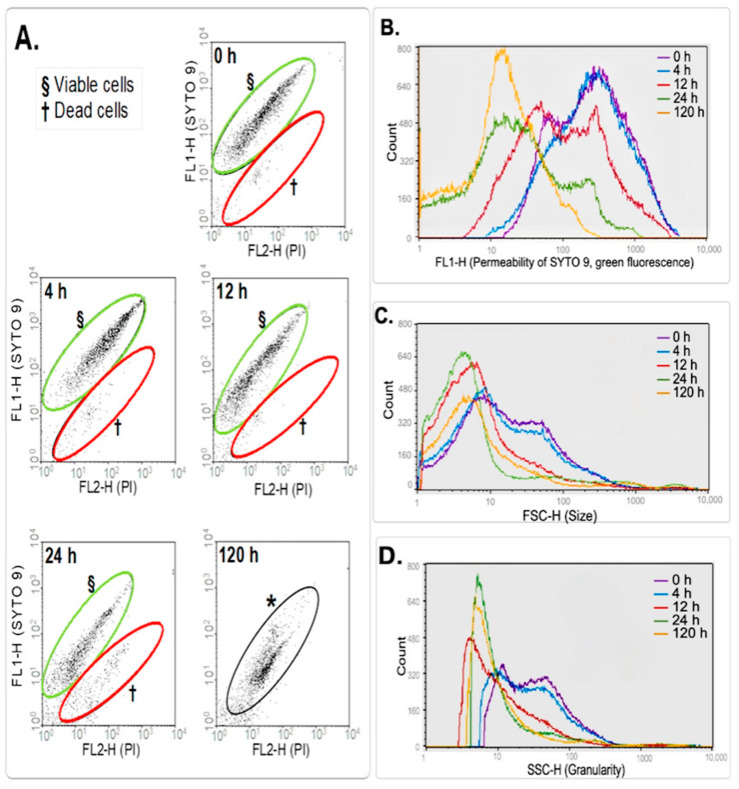
Multiparametric FCM analysis of *M. smegmatis* grown under starvation conditions. (**A**) Dot plots of dead (†), viable (§) and the combination of dead and viable (*) cell populations are displayed; green fluorescence (FL1-H, SYTO 9) vs. red fluorescence (FL2-H, PI) is charted through time (0 h to 120 h). (**B**) Overlap of frequency histograms of the fluorescence profile of SYTO 9 (FL1-H) at different starvation times. (**C**) Histograms of cell size. The forward scatter depicts size (FSC-H), and the ordinate (cell counts) denotes single events. (**D**) Frequency histograms of cell granularity/complexity (SSC-H); the ordinate (cell counts) denotes single events.

**Figure 3 microorganisms-12-02280-f003:**
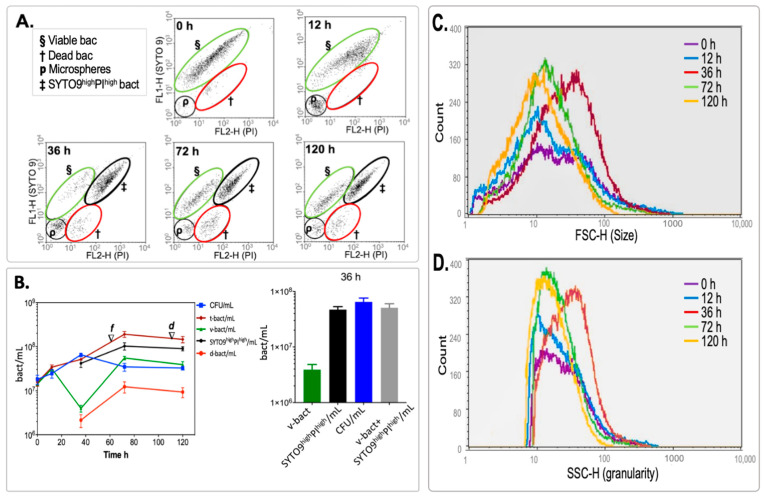
Multiparametric analysis of the results obtained by FCM of M. smegmatis in hypoxia. (**A**) Dot plots of dead cells defined as SYTO 9lowPIhigh (†), viable cells defined as SYTO 9^high^PI^low^ (§) and cells defined as SYTO 9^high^PI^high^ (‡) are displayed; fluorescence microspheres were used for calibration (p), red fluorescence (FL2-H, PI) vs. green fluorescence (FL1-H, SYTO 9) is charted through time (0 h to 120 h). (**B**) Measurements of mycobacterial populations detected by FCM and their correlation with measurements in solid media. The blue squares represent CFU/mL; brown diamonds represent total bacteria per mL (t-bact/mL); black circles represent SYTO 9^high^PI^high^ bacteria/mL (SYTO 9^high^PI^high^-bact/mL); green triangles represent viable bacteria per mL (v-bact/mL); red circles represent dead bacteria per mL (d-bact/mL). The times when a noticeable fading (NRP1, ▽f) and the total discolouration (NRP2, ▽d) of the methylene blue occurred are indicated. Values are displayed on a logarithmic scale and mean ± SD are charted. All experiments were carried out in triplicate. (**C**,**D**) Frequency histograms of cell size (FSC-H) and cell complexity (SSC-H) of *M. smegmatis*, respectively. In both cases, the ordinate (cell counts) denotes single events.

**Figure 4 microorganisms-12-02280-f004:**
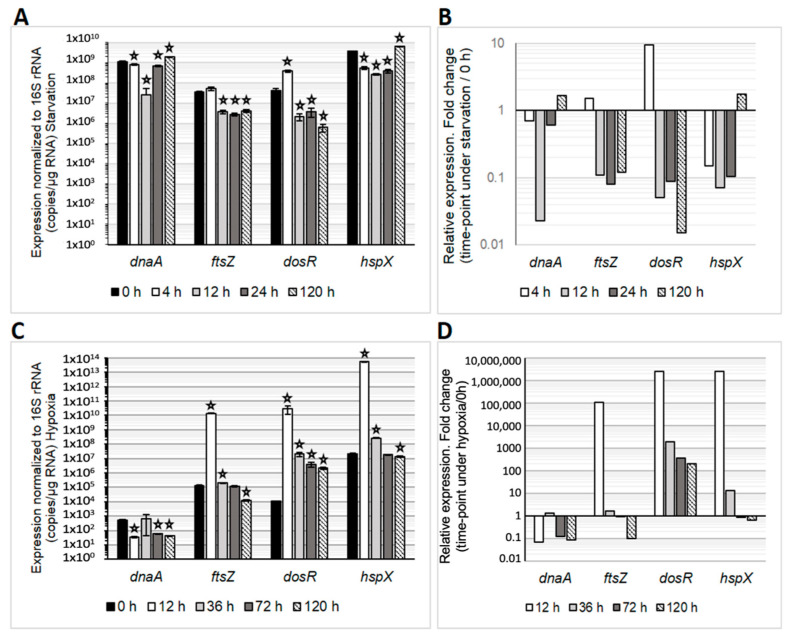
Gene expression of *M. smegmatis* cultures under in vitro latency conditions over time. Absolute gene expression normalised to 16S rRNA was measured in *M. smegmatis* either under starvation (**A**) or hypoxia (**C**); standard deviations are charted and *p* < 0.05 was considered significantly different (represented by a star) between expression in time points of in vitro latency and the time 0 h (exponential phase). The relative gene quantification in starvation (at 4, 12, 24 and 120 h) is expressed as the ratio of transcription overtime/transcription at 0 h (**B**). The relative gene quantification in hypoxia (at 12, 36, 72 and 120 h) is expressed as the ratio of transcription overtime/transcription at 0 h (**D**).

**Figure 5 microorganisms-12-02280-f005:**
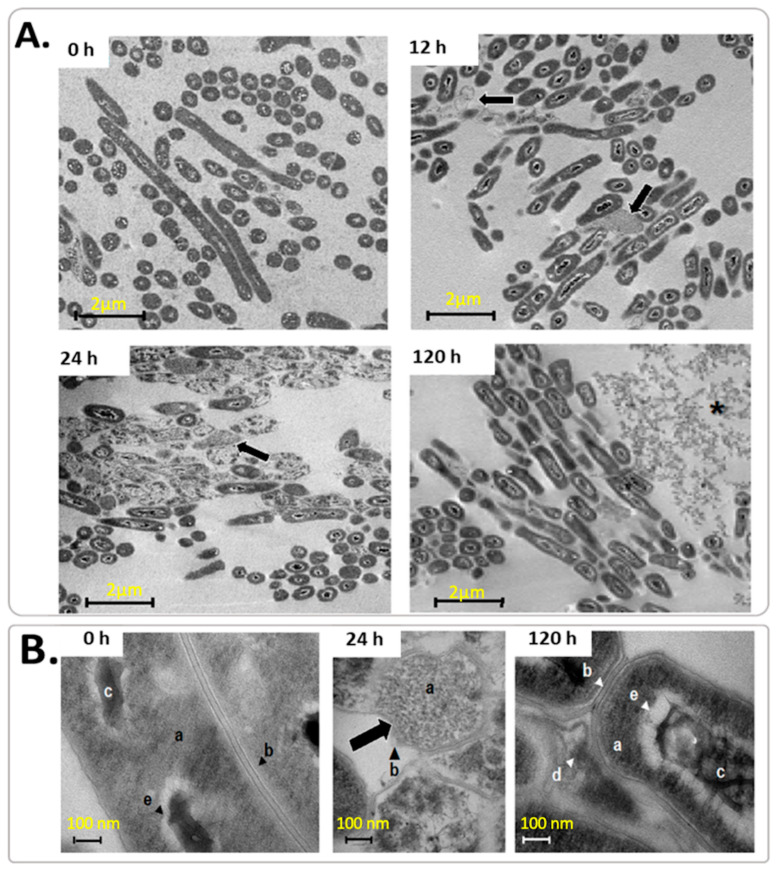
TEM images of *M. smegmatis* under starvation. (**A**) Representative electron micrographs at 0 h, 12 h, 24 h, and 120 h of incubation; magnification was ×10 K; scale bars represent 2 µm; arrows point to electron-transparent cells without nucleoid and debris are indicated with an asterisk. (**B**) Ultrastructure of *M. smegmatis* at 0 h, 24 h, and 120 h of incubation; magnification was ×150 K and scale bars represent 100 nm. Arrow heads point to different cells structure such as a a: cytoplasmatic material, b: cell wall, c: chromatin, d: cellular debris, and e: electron-transparent zone surrounding the nucleoid. The acceleration voltage was 70 kV. These images represent three independent experiments and were chosen from five random fields.

**Figure 6 microorganisms-12-02280-f006:**
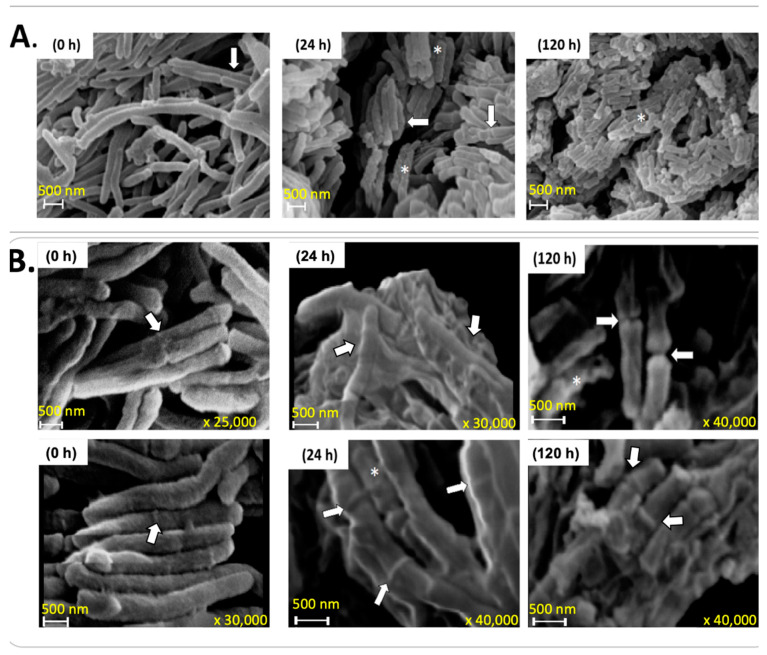
SEM images of *M. smegmatis* under starvation. (**A**) Cells at 0, 24 and 120 h of starvation, at ×10 K magnification. (**B**) Cells at ×25 K, ×30 K and ×40 K magnification at 24 and 120 h of starvation. Rod cell surfaces with a rough appearance are indicated with an asterisk, and septa are indicated by an arrow. These images represent three independent experiments and were chosen from five random fields. Scale bars represent 500 nm, and the acceleration voltage was 15 kV in all cases.

**Figure 7 microorganisms-12-02280-f007:**
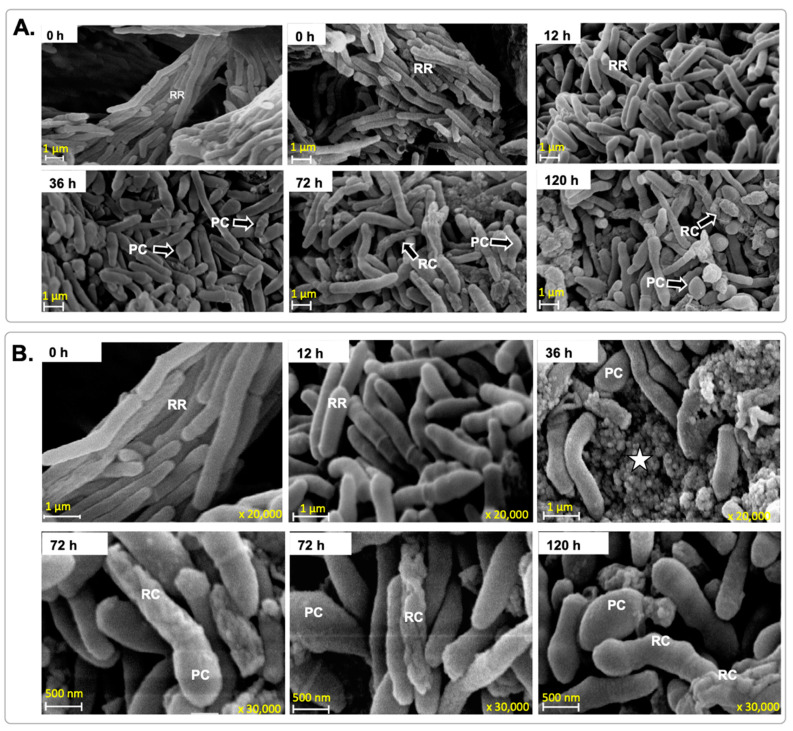
SEM images of *M. smegmatis* cells grown under hypoxic conditions. (**A**) Representative electron micrographs taken at various times; magnification was ×10 K, and scale bars represent 1 µm. (**B**) Morphological changes at a higher magnification; times 0 to 36 h were ×20 K, and the scale bar represents 1 µm; times 72 to 120 were ×30 K, and scale bars represent 500 nm. Regular rods (RR), pleomorphic cells (PC), and rough surface cells (RC); the star indicates cellular debris. The acceleration voltage was 15 kV. These images represent three independent experiments and were chosen from five random fields.

**Figure 8 microorganisms-12-02280-f008:**
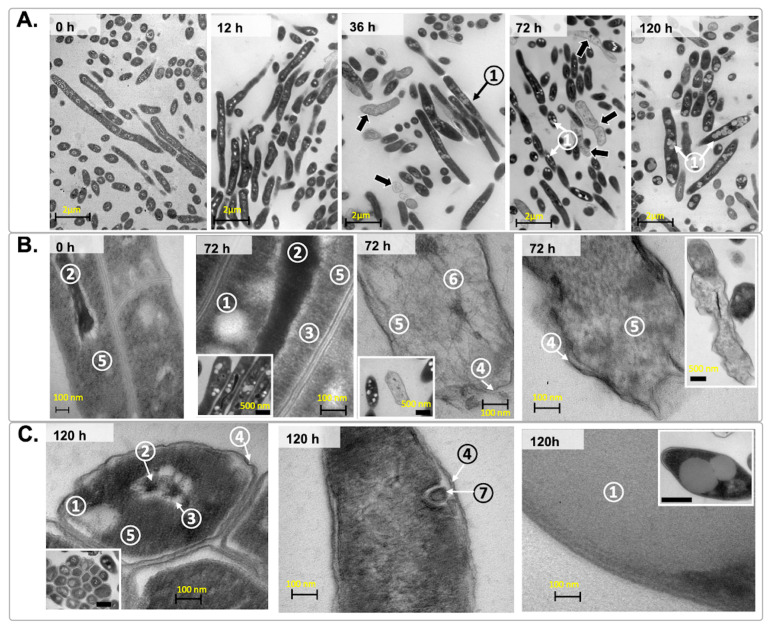
TEM images of *M. smegmatis* under hypoxia. (**A**) *M. smegmatis* cells over time, at ×10K magnification. Scale bars represent 2 µm. (**B**) *M. smegmatis* cells at 0 h and 72 h under hypoxia, magnification was ×45 K or ×150 K, respectively. Scale bars represent 100 nm. (**C**) *M. smegmatis* cells at 120 h under hypoxia, magnification was ×150 K. Scale bars represent 100 nm. Black arrows point to cells with a low density of electrons. Numbers in panels point to ① Lipid-body-like structures, ② compacted DNA with highly dense electrons, ③ low-density electron zone, ④ loss of rigidity of cell wall, ⑤ cytoplasm, ⑥ star-shaped nucleoid and ⑦ vesicular structures with internal membranes. An acceleration voltage of 70 kV was used. Smaller frames within each panel were magnified ×120 K, and the acceleration voltage was 60 kV; scale bars represent 500 nm. These images represent three independent experiments and were chosen from five random fields.

**Table 1 microorganisms-12-02280-t001:** Effect of starvation on morphological properties of *M. smegmatis* mc^2^ 155.

Period of Cell Culture Under Starvation (h)	Cell Length (µm) ^§^	Morphology ^‡^
* 0	3.83 ± 1.29	Rods
4	2.90 ± 0.79	Rods
12	2.11 ± 0.28	Rods
24	1.52 ± 0.34	Rods with rough appearance
120	1.41 ± 0.34	All rods have a rough surface

* Culture at exponential phase. ^§^ Cell sizes were determined from transmission electron micrographs, and at least 50 cells were measured; changes were principally in length; diameter was essentially unchanged. ^‡^ Morphology was determined from scanning electron micrographs; more than five fields of view were analysed.

**Table 2 microorganisms-12-02280-t002:** Effect of hypoxia on morphological properties of *M. smegmatis*.

Period Under Hypoxia	Cell Length (µm) ^§^	Morphology ^‡^ (% of Fraction of Cells)			Pleomorphic + Rough Cells (%)
		Rod-Shaped	Pleomorphic ^▲^	Rough	
* 0 h	3.83 ± 1.29	100	0	0	0
12 h	3.04 ± 0.91	75	25	0	25
36 h	3.73 ± 1.42	16	79	5	84
72 h	3.53 ± 1.76	18	67	15	82
120 h	3.38 ± 0.73	16	56	28	84

* Culture in exponential phase. ^§^ Cell sizes were determined from transmission electron micrographs; for rod-shaped cells, at least 50 cells of normal morphology were measured. ^‡^ Morphology was determined from scanning electron micrographs; more than five random fields of view were analysed. ^▲^ Variants of rods.

## Data Availability

The original contributions presented in the study are included in the article/Appendix A, further inquiries can be directed to the corresponding author.

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
