# Peer review of "Novel Populations of Mycobacterium smegmatis Under Hypoxia and Starvation: Some Insights on Cell Viability and Morphological Changes"

_microorganisms, 2024, doi:10.3390/microorganisms12112280_

Round 1
Reviewer 1 Report
Comments and Suggestions for Authors
Comments to M. smegmatis paper
The manuscript describes features of Mycobacterium smegmatis in the dormant state after exposure to harsh conditions such as hypoxia and starvation. Most of the data are based on interpretation of flow cytometry data of SYTO 9/PI staining, complemented by SEM and TEM images and CFU. There are some concepts that have to be corrected as stated below.
Abstract:
Line 27: The authors state that the dormant cells lack a nucleus: "anucleated" small cells. But bacteria do not have a nucleus, such that another descriptive word should be used.
Line 29: The abbreviation NRP1 should be spelled out the first time mentioned.
Introduction:
Line 42: Instead of Mtb, use the classical abbreviation of M. tuberculosis.
Is VBNC a laboratory-failure concept of cultivation?
Method section:
Line 74: Are you sure that 2 should be in superscript? Usually, this strain is written as MC2. Please check and correct. The same for line 182 in Result section.
Line 75: The abbreviations BBL and ADC should be defined, as well as describing their compositions.
Line 89: The source and composition of the Dubos medium should be presented.
Line 90: The Wayne and Hayes model should be described. And the definition of replicative persistence 1 and non-replicative 2 phases should be defined.
Section 2.5: Line 119: Are you sure that 530 is the excitation and not the emission? Usually, excitation is with the 488nm laser. SYTO 9 fluorescence leaks into the 585nm channel, such that PI should be captured in dsRed channel instead. It should be stressed that SYTO 9 stains the nucleic acid of both live and dead cells, while PI only enters bacteria with comprised membrane.
The exact method of determining the total number of bacteria on flow cytometry should be described.
Line 150: The total RNA isolation should be described in more detail.
Section 2.9: 16S rRNA is not a good house-keeping gene. Additional house-keeping genes should be used for normalization. The primer sequences should be provided.
Result section.
Figure 1B lacks the green curve. Although argumented in the text later, it ought to be presented in this figure.
Obscure abbreviations such as v-bact and t-bact should be spelled out.
Line 233: The "altruistic suicide" is a nice idea, but does not seem to be the case. The SYTO 9/PI staining depends on the bacterial growth conditions preceding the analysis. Were the bacteria taken from an overnight culture, or after an initial growth phase? This has to be described and mentioned in the text. The dilution factor and incubation time for reaching the exponential phase should be mentioned.
Figure 2A: These are dot plots (not cytograms). Please correct the text.
Figures 2A and B and C should include unstained bacteria. The loss of SYTO 9 staining after 24 and 120 h is an indication for dead bacteria which have lost DNA content. It is sufficient to say "histograms", no need to write "overlap of frequencies". Histograms of PI should be shown. A summary bar graph should be presented showing the percentage of PI positive bacteria over time. Moreover, changes in the geometric mean (rather than the "peak") should be summarized in a graph for the different parameters.
Lines 246-247 – the geometric mean which represents the entire population rather than the peak which represent a single maximum event should be presented. This is especially important for histograms having more than one peak.
Line 257-259 – The size of the bacteria can not be used as a criterion for cell division. BrDU staining can be used for such purposes.
Line 265: Anucleated cells – can not be used for bacteria. It is also not clear how this is related to FtsZ and DnaA expression.
Line 270: Unknown cells – should be better characterized – There are SYTO 9highPIhigh cells. These bacteria are an intermediate stage between live and dead bacteria – after membrane perforation, but before cytoplasmic leakage of SYTO 9. The latter (SYTO9lowPIhigh) represents the dead bacteria. Such bacterial populations have been previously described by other research groups. Thus, you can "define" the "unknown".
Fixation of bacteria followed by DAPI staining or Hoechst staining may provide information of the DNA content per bacteria. Such an assay is important to complement the SYTO 9/PI staining.
Line 275: Please add an h after 120.
Since the medium is not changed during the 120 h incubation period (what was the incubation volume?), bacterial cell death is likely caused by nutrient depletion.
Lines 28-286: Does the presence of MB affect bacterial viability? (e.g.., https://doi.org/10.3390/ph17020241).
Line 293: The authors claim that the SYTO 9highPIhigh cell population are cells well adapted to hypoxic conditions. To prove this, the different SYTO9/PI populations should be separated by FACS, and then allowed to recover.
Figure 3A: Correct to "Dot plots". Correct "unknown" to SYTOhighPIhigh; viable should be defined as SYTOhighPIlow and dead defined as SYTO9lowPIhigh. P in the dot plots is not defined in the legend. These are dead cells which have lost nucleic acid (SYTO9lowPIlow).
Figure 3B: The total bacteria are in brown, and not in red color as described in the legend. Please correct.
Line 321: Bacteria do not have nuclei. So here is a misinterpretation.
Line 338: MCC should be written in full name.
Line 340: You can't say "new-born". Maybe better to say, recently divided cells.
Figure 5: A higher magnification of time 0 should also be presented. According to Figure 5B, the bacteria seem to be held together in chains. Is it so? More images should be provided (three images of each time point).
Lines 348-352: The gene expression studies should be better described.
The supplementary figures can be added to the main text.
Lines 376-383: The assumption that the pleomorphic cells are "u-bact" has no basis. Again, this issue can be solved by sorting for the u-bact (SYTOhighPIhigh) population which is then visualized by SEM. mRNA profiling can also be done on the bacteria subpopulations obtained by FACS.
The discussion should be corrected according to the new data that will be obtained.
Author Response
The manuscript describes features of Mycobacterium smegmatis in the dormant state after exposure to harsh conditions such as hypoxia and starvation. Most of the data are based on interpretation of flow cytometry data of SYTO 9/PI staining, complemented by SEM and TEM images and CFU. There are some concepts that have to be corrected as stated below.
Abstract:
Line 27: The authors state that the dormant cells lack a nucleus: "anucleated" small cells. But bacteria do not have a nucleus, such that another descriptive word should be used.
Reply: Bacterial cells without nucleoid are called anucleated cells (Mishra et al., 2021; PMID: 33427942; Mäkelä et al., 2021; PMID: 34385314; Pradhan et al., Oyamada et al., 2006; PMID: 16377708). Now this term is mentioned in line 305 “anucleated cells (cells without nucleoid)”
Line 29: The abbreviation NRP1 should be spelled out the first time mentioned
Reply: Nonreplicating persistence 1 (NRP1) is now included. In the text line 29
Introduction:
Line 42: Instead of Mtb, use the classical abbreviation of M. tuberculosis.
Reply: We included the classical abbreviation of M. tuberculosis in the manuscript
Is VBNC a laboratory-failure concept of cultivation?
Reply: It is not a failed concept; it is a group of bacteria that are in an inactive or dormant state because they grew in unfavorable conditions. These cells require the presence of reactivation-promoting proteins (Rpf, Resuscitation promoting factor) for the reactivation of the growth of these cells to occur.
Method section:
Line 74: Are you sure that 2 should be in superscript? Usually, this strain is written as MC2. Please check and correct. The same for line 182 in Result section.
Reply: We apologize for this, now this change is amended as “mc2” according to ATTC nomenclature
Line 75: The abbreviations BBL and ADC should be defined, as well as describing their compositions.
Reply BBL is a registered trademark of Becton Dickinson. In the text line 76. The wrong supplement was named previously the proper correction can be found here: ADC supplement: 0.2% (w/v) dextrose, 0.2% (v/v) glycerol, 0.5% bovine serum albumin, catalase (4 mg/L) and 15 mM sodium chloride (BD, USA).
Line 89: The source and composition of the Dubos medium should be presented.
Reply: In lines 74-76 the composition and source of Dubos medium is mentioned, which will be mentioned continuously from then on.
Line 90: The Wayne and Hayes model should be described. And the definition of replicative persistence 1 and non-replicative 2 phases should be defined.
Reply: The Wayne and Hayes model as well as the non-replicative persistence 1 and non-replicative 2 phases have been widely described in previous works (line 94); references are attached below for consultation.
Wayne, L.G.; Hayes, L.G. An in vitro model for sequential study of shiftdown of Mycobacterium tuberculosis through two stages of nonreplicating persistence. Infect. Immun. 1996, 64, 2062–2069.
Badillo-Lopez, C.; Gonzalez-Mejia, A.; Helguera-Repetto, A.C.; Salas-Rangel, L.P.; Rivera-Gutierrez, S.; Cerna-Cortes, J.F.; Gonzalez, Y.M.J.A. Differential expression of dnaA and dosR genes among members of the Mycobacterium tuberculosis complex under oxic and hypoxic conditions. Int. Microbiol. 2010, 13, 9–13.
Soto-Ramirez, M.D.; Aguilar-Ayala, D.A.; Garcia-Morales, L.; Rodriguez-Peredo, S.M.; Badillo-Lopez, C.; Rios-Muniz, D.E.; MezaSegura, M.A.; Rivera-Morales, G.Y.; Leon-Solis, L.; Cerna-Cortes, J.F.; et al. Cholesterol plays a larger role during Mycobacterium tuberculosis in vitro dormancy and reactivation than previously suspected. Tuberculosis 2017, 103, 1–9.
Section 2.5: Line 119: Are you sure that 530 is the excitation and not the emission? Usually, excitation is with the 488nm laser. SYTO 9 fluorescence leaks into the 585nm channel, such that PI should be captured in dsRed channel instead. It should be stressed that SYTO 9 stains the nucleic acid of both live and dead cells, while PI only enters bacteria with comprised membrane.
Reply: We apologize for this; the change has been made to “excitation at 488 ± 15 nm” (line 137).
The exact method of determining the total number of bacteria on flow cytometry should be described.
Reply: This information is now provided in lines 115 to 128.
​
Line 150: The total RNA isolation should be described in more detail.
Reply: As suggested by the reviewer we included this description in the text (lines 168-176)
Section 2.9 16S rRNA is not a good house-keeping gene. Additional house-keeping genes should be used for normalization. The primer sequences should be provided.
Reply: The 16S rRNA gene has been used in previous work in our laboratory, with satisfactory results (26). Primer sequences were: 16SF: 5′ATGACGGCCTTCGGGTTGTAA-3′ and 16SR: 5′-CGGCTGCTGGCACGTAGTTG-3′. In the text lines 184-186
Result section.
Figure 1B lacks the green curve. Although argumented in the text later, it ought to be presented in this figure. Obscure abbreviations such as v-bact and t-bact should be spelled out.
Reply: The green curve corresponding to viable-bact/mL was included in the Figure 1B, regarding to abbreviations both are spelled out in the legend of this figure as well as in the text lines 219 and 220.
Line 233: The "altruistic suicide" is a nice idea but does not seem to be the case. The SYTO 9/PI staining depends on the bacterial growth conditions preceding the analysis. Were the bacteria taken from an overnight culture, or after an initial growth phase? This has to be described and mentioned in the text. The dilution factor and incubation time for reaching the exponential phase should be mentioned.
Reply: In our initial hypothesis, we considered that live bacteria might utilize nutrients from dead or compromised cells to support their survival during starvation. However, upon further reflection and in agreement with reviewer feedback, we acknowledge that our current data do not substantiate this claim. Our observations indicate that after 12 hours of starvation, the mycobacterial population fails to sustain growth, leading to a state where, by 120 hours, the entire population exhibits propidium iodide (PI) staining. This uniform PI staining suggests widespread membrane compromise, indicating a decline in cell viability and pointing towards an inevitable fatal outcome for the bacterial population under these conditions.
As described in the methods section, the starvation model began with a culture in the logarithmic growth phase, where the bacterial population was assumed to be viable, as illustrated in Figure 2A (0 hours of the starvation model). At this initial time point, the bacterial cells were in optimal conditions for growth, confirming their viability at the start of the experiment. As also mentioned in this section the exponential growth phase was reached, at 20 h of incubation with an Optical Density at 640 nm (OD640) equal to 1.0 ± 0.05. From this culture at exponential phase, 100 mL of M. smegmatis cultures were centrifuged and resuspended in 100 mL sterile phosphate-buffered saline (PBS).
Figure 2A: These are dot plots (not cytograms). Please correct the text.
Reply: This change is amended as “dot plots” in the manuscript
Figures 2A and B and C should include unstained bacteria. The loss of SYTO 9 staining after 24 and 120 h is an indication for dead bacteria which have lost DNA content. It is sufficient to say "histograms", no need to write "overlap of frequencies". Histograms of PI should be shown.
Reply: The aim of this staining and corresponding figure is to demonstrate the loss of SYTO 9 staining, which marks viable bacteria. As the SYTO 9 signal diminishes, the increase in PI staining indicates a loss of viability, membrane compromise, and a likely fatal outcome if the bacteria remain in this stress condition. Therefore, we believe that no additional data are needed, as they could introduce confounding variables. Figure 2A clearly shows that the bacteria lose their membrane integrity, allowing PI to disseminate and stain the DNA. Figure 2B illustrates the progressive fading of viability staining, while Figure 2C shows the reduction in bacterial size. Together, these figures effectively capture the detrimental effects of radical starvation on the bacteria.
A summary bar graph should be presented showing the percentage of PI positive bacteria over time. Moreover, changes in the geometric mean (rather than the "peak") should be summarized in a graph for the different parameters.
Reply: As the reviewer suggested, we included a bar graph as Supplementary Figure S1, that shows the percentage of PI positive bacteria over time.
Lines 246-247 – the geometric mean which represents the entire population rather than the peak which represent a single maximum event should be presented. This is especially important for histograms having more than one peak.
Reply: This could also be explained in Supplementary Figure S1, which shows the percentage of total population-stained whit SYTO9 or PI.
Line 257-259 – The size of the bacteria can not be used as a criterion for cell division. BrDU staining can be used for such purposes.
Reply: We added more information about how ftsZ and dnaA expression could also be employed to explain cell division process. Now, it is on lines 299-311.
Line 265: Anucleated cells – can not be used for bacteria. It is also not clear how this is related to FtsZ and DnaA expression.
Reply: Bacterial cells without nucleoid are called anucleated cells (Mishra et al., 2021; PMID: 33427942; Mäkelä et al., 2021; PMID: 34385314; Pradhan et al., Oyamada et al., 2006; PMID: 16377708). To clarify how the presence of anucleated cells is related to ftsZ and dnaA expression, we have corrected and added the paragraph on previous line 265. Now, it is on lines 299-312. We thank the reviewer for this important observation.
Line 270: Unknown cells – should be better characterized – There are SYTO 9highPIhigh cells. These bacteria are an intermediate stage between live and dead bacteria – after membrane perforation, but before cytoplasmic leakage of SYTO 9. The latter (SYTO9lowPIhigh) represents the dead bacteria. Such bacterial populations have been previously described by other research groups. Thus, you can "define" the "unknown".
Reply: Unknown cells in effect are SYTO 9highPIhigh, therefore we changed this term, stating that they are in the intermediate stage between live and death bacteria (line 324).
Fixation of bacteria followed by DAPI staining or Hoechst staining may provide information of the DNA content per bacteria. Such an assay is important to complement the SYTO 9/PI staining.
Reply: We did not perform another kind of staining because the primary aim of the study is to evaluate bacterial viability and membrane integrity using the SYTO 9/PI staining method. These dyes specifically differentiate live and dead bacteria based on membrane integrity, which is critical for understanding the effects of starvation on mycobacterial populations.
Indeed, DAPI and Hoechst staining provide information about DNA content but do not necessarily add significant insights into viability beyond what SYTO 9/PI already offers; also, they do not differentiate between viable and non-viable cells effectively. In contrast, SYTO 9/PI staining directly measures membrane integrity, a more critical indicator of cell viability in this context. By focusing solely on SYTO 9/PI staining, the study maintains a clearer narrative regarding the effects of starvation on mycobacterial viability. Adding further methods could introduce confusion or dilute the emphasis on the main findings.
Line 275: Please add an h after 120.
Reply: We amended this.
Since the medium is not changed during the 120 h incubation period (what was the incubation volume?), bacterial cell death is likely caused by nutrient depletion.
Reply: The incubation volume of 100 mL was described in the material and methods section. We believe that cell death results from the inability of mycobacteria to adapt to extreme starvation conditions. This model involves subjecting the bacteria to starvation without an adaptation period, in contrast to the gradual onset of hypoxia seen in the Wayne model. In our starvation model, all carbon sources are removed from the beginning, whereas hypoxia develops more slowly.
25.- Lines 28-286: Does the presence of MB affect bacterial viability? (e.g.., https://doi.org/10.3390/ph17020241).
Reply: In this study, we used parallel cultures with methylene blue (MB) at a concentration of 1.5 µg/mL as an indicator of oxygen depletion, which was used only to establish the time required to reach each NRP phase. All experimental determinations were performed using cultures without MB.
Line 293: The authors claim that the SYTO 9highPIhigh cell population are cells well adapted to hypoxic conditions. To prove this, the different SYTO9/PI populations should be separated by FACS, and then allowed to recover.
Reply: As shown in Figure 3B, we quantified viable bacteria and those adapted to hypoxia using CFU/mL measurements. The data indicate that at 36 hours in the Wayne model, the bacteria are capable of multiplying similarly to traditionally viable bacteria. This suggests that adaptation to hypoxia leads to a population of viable bacteria exhibiting different permeability characteristics, as indicated by the staining results. At this 36-hour mark under hypoxic conditions, the adapted population, identified by SYTO 9high and PIhigh staining, is larger than the classical viable and dead bacterial populations.
Figure 3A: Correct to "Dot plots". Correct "unknown" to SYTOhighPIhigh; viable should be defined as SYTOhighPIlow and dead defined as SYTO9lowPIhigh. P in the dot plots is not defined in the legend. These are dead cells which have lost nucleic acid (SYTO9lowPIlow).
Reply: We have improved Figure 3A by changing it to dot plots and removing the term “unknow” to SYTOhighPIhigh, both in the figure and the manuscript text. The relationship between the SYTO and PI intensities in viable and dead bacterial populations is now indicated in the caption of this figure as well as in the main text.
The abbreviation “P” was mentioned in the initial submission of the manuscript to indicate the presence of fluorescence microspheres (p) used for calibration; these are not cells.
Figure 3B: The total bacteria are in brown, and not in red color as described in the legend. Please correct.
Reply: We have amended this
Line 321: Bacteria do not have nuclei. So here is a misinterpretation.
Reply: Bacterial cells without nucleoid are called anucleated cells (Mishra et al., 2021; PMID: 33427942; Mäkelä et al., 2021; PMID: 34385314; Pradhan et al., Oyamada et al., 2006; PMID: 16377708). Now this term is mentioned in line 305 “anucleated cells (cells without nucleoid)”
Line 338: MCC should be written in full name.
Reply: The abbreviation of MCC was spelled out the first time mentioned in line 244, after that we only use this abbreviation.
Line 340: You can't say "new-born". Maybe better to say, recently divided cells.
Reply: We amended this
Figure 5: A higher magnification of time 0 should also be presented. According to Figure 5B, the bacteria seem to be held together in chains. Is it so? More images should be provided (three images of each time point).
Reply: As the author kindly suggested, we included a higher magnification of time 0. Regarding figure 5B, the bacilli have many septa and is observed like chains at 24h after that a reduction in cell length is informed, now more images are provided.
Lines 348-352: The gene expression studies should be better described.
Reply: As mentioned above extra information about how ftsZ and dnaA expression (lines 299-312)
The supplementary figures can be added to the main text.
Reply: As the author kindly suggested, we included the supplementary figures to the main text.
Lines 376-383: The assumption that the pleomorphic cells are "u-bact" has no basis. Again, this issue can be solved by sorting for the u-bact (SYTOhighPIhigh) population which is then visualized by SEM. mRNA profiling can also be done on the bacteria subpopulations obtained by FACS.
Reply: We redefined u-bact as SYTOhighPIhigh-bact population and agree with the reviewers' comments that further analyses are required to characterize these cells. We have now included this perspective in the discussion section lines 542 to 547.
The discussion should be corrected according to the new data that will be obtained.
Reply: We improved our discussion according to the reviewers' suggestions.
Reviewer 2 Report
Comments and Suggestions for Authors
It is an intriguing study on the subpopulations of bacteria during starvation and hypoxia. The experiments are well designed and performed. However, I have several significant questions about the interpretation of the data.
1. Lines 200-202
The number of viable cells was higher than CFU between 4 h and 12 h of starvation. The authors supposed the presence of viable non-culturable cells, which is indeed a good explanation. However, could it be caused by cell clumping, which is also a known phenomenon? Could this possibility be omitted based on the experiments and observations? The authors used Tween and shaking for the cultures, but I am not sure it is enough.
The same question is relevant for the explanation of the increase in OD and the decrease in CFU (lines 413-416). Cell clumping could lead to an increase in OD.
2. Figure 3
I suppose that the cluster analysis of viable cells at 12 h is not exactly correct. We could see the clear difference in cytograms of viable cells between 0 h and 12 h, but I suppose the top right part is the same as ‘Unknown’ cells at 36 h. At 12 h, the ‘viable’ and ‘unknown’ overlap partially. Thus, this ‘unknown’ population appeared before the noticeable fading of Methylene Blue.
3. Line 443 and below
The authors discuss the limitations of PI staining as specific ‘membrane-compromised’ cells. However, it seems that SYTO 9 also has limitations. As stated in the manual, 'When used alone, the SYTO 9 stain generally labels all bacteria in a population - those with intact membranes and those with damaged membranes.’ Thus, it seems to be an unspecific DNA binding agent and, therefore, the population stained with SYTO 9 should be treated as ‘dead or alive with chromosome' and not as ‘viable’.
4. Does the widely discussed in the manuscript ‘membrane-compromised’ cells be the same phenomenon as L-form cells? It is an old days conception of ‘filtered’ and ‘L-forms’, see, for example, Markova, 2012, PMID: 22495116.
5. The authors missed one publication about Large Resting Cells and Small Resting Cell in M. smegmatis – Wu et al. PMID: 27784279. It is highly desirable to review the results in light of this study.
Author Response
Point by point response to the reviewer comments:
Reviewer 2
It is an intriguing study on the subpopulations of bacteria during starvation and hypoxia. The experiments are well designed and performed. However, I have several significant questions about the interpretation of the data.
1.- Lines 200-202 The number of viable cells was higher than CFU between 4 h and 12 h of starvation. The authors supposed the presence of viable non-culturable cells, which is indeed a good explanation. However, could it be caused by cell clumping, which is also a known phenomenon? Could this possibility be omitted based on the experiments and observations? The authors used Tween and shaking for the cultures, but I am not sure it is enough.
Reply: As you mentioned, the discrepancies between the first two determinations during the starvation could be due to the presence of large cell aggregates in the cultures, but these aggregates are broken up during the determination of the UFC/mL, because we homogenize the aliquot by using glass beads (710 -1180 mm) before seeding on the agar plates.
On the other hand, the bacterial cultures were grown in the presence of Tween, and the flask contained a stainless-steel spring to prevent the formation of cell aggregates.
The same question is relevant for the explanation of the increase in OD and the decrease in CFU (lines 413-416). Cell clumping could lead to an increase in OD.
Reply: These changes in the size of M. smegmatis cells could correspond to an adaptive strategy called “reductive cell division”, this phenomenon has been studied in Gram-negative bacteria (Wanner & Egli, 1990). Reductive cell division could largely explain why during the first 12 h of incubation of M. smegmatis cultures in lack of nutrients, an increase in OD is observed even though there is no increase in the number of in CFU, the cells divide, giving rise to anucleate cells, which increases the density of the cultures, and therefore increases the amount of light scattered by them.
Wanner U. & Egli T. (1990). Dynamics of microbial growth and cell comosition in batch culture. FEMS Microbiol. Rev. 6:19-43.
2.- Figure 3. I suppose that the cluster analysis of viable cells at 12 h is not exactly correct. We could see the clear difference in cytograms of viable cells between 0 h and 12 h, but I suppose the top right part is the same as ‘Unknown’ cells at 36 h. At 12 h, the ‘viable’ and ‘unknown’ overlap partially. Thus, this ‘unknown’ population appeared before the noticeable fading of Methylene Blue.
Reply: The reviewer’s observation is accurate. At 12 hours, we notice that viable cells, similar to those observed at 0 hours, begin to show changes in their staining properties. However, the most significant changes become evident after 36 hours in the hypoxia model. Due to the hermetically sealed environment of the model, there is no air exchange in the medium, causing bacterial metabolism to deplete oxygen from the start. As a result, adaptation to the stress conditions occurs gradually. Therefore, we chose 36 hours as the clear point at which distinct staining populations can be observed. Lines 293-298.
3.- Line 443 and below. The authors discuss the limitations of PI staining as specific ‘membrane-compromised’ cells. However, it seems that SYTO 9 also has limitations. As stated in the manual, 'When used alone, the SYTO 9 stain generally labels all bacteria in a population - those with intact membranes and those with damaged membranes.’ Thus, it seems to be an unspecific DNA binding agent and, therefore, the population stained with SYTO 9 should be treated as ‘dead or alive with chromosome' and not as ‘viable’.
Reply: We defined the populations based on SYTOhighPIhigh-bact depending on the fluorescence level, high SYTO High or low SYTO Low, we decided to define the populations based on their staining capacities due to the confusion that would arise from the population of non-culturable bacteria that could be confused with dead ones.
4.- Does the widely discussed in the manuscript ‘membrane-compromised’ cells be the same phenomenon as L-form cells? It is an old days conception of ‘filtered’ and ‘L-forms’, see, for example, Markova, 2012, PMID: 22495116.
Reply: The ‘‘membrane-compromised cells’ were described according to FCM experiments, we named this cell population exhibited the same intensity of fluorescence coming from the two fluorochromes used in FCM (SYTOhighPIhigh), Although agrees with some aspects in the definition described by Markova in M. tuberculosis and M. bovis BCG for L-forms we have to demonstrate if this population can growth in fresh culture media and form colonies isolating this population. Now this information in commented in lines 506 and 507.
5.- The authors missed one publication about Large Resting Cells and Small Resting Cell in M. smegmatis – Wu et al. PMID: 27784279. It is highly desirable to review the results in light of this study.
Reply: We would like to thank for this kindly suggestion of the reviewer, now we compared our data with this article and mentioned in the discussion section lines: 510 to 519.
Round 2
Reviewer 1 Report
Comments and Suggestions for Authors
Comments to revised microorganisms-3242965
The manuscript has been improved. However, there are several issues that still need to be addressed before the manuscript can be accepted for publication.
Line 93: The definition of the two dormant phases NRP1 and NRP2 should be stated.
Line 100: Please add space after the parenthesis.
Lines 115-116 are a repetition of previous sentence. Please delete one of them.
Line 119-120: The final concentration of SYTO 9 and PI should be stated.
Line 121: The composition of Kohn-Harris medium should be stated.
Sentence in line 167 is incomplete (lacks a verb).
Line 174: I do not think you washed the RNA with Trizol reagent, it should be washed twice with 70% ethanol. Please check the accuracy of the text.
Section 2.9: The sequence of all primers should be presented.
Line 210: The morphological changes are not shown in Figure 1. Please show them and describe which morphological changes have occurred.
Figure 1 lacks the A and B labeling in the figure itself. I would also suggest adding "starvation" and "hypoxia" above the respective subfigures.
Figure 1B, green graph at around 36 h, the stdev is only in one direction (minus), the plus stdev is lacking.
Lines 224- : It is important to refer in the text to Figure 1A.
Line 238: You have to describe in words the difference observed, and the statistical significance has to be added. Statistical significance should also be indicated in Figure 1A and B. How many times was Figure 1B repeated? This should be stated in the figure legend. Is this drop in SYTO 9 positive cells seen in all experiments of hypoxia?
Section in lines 240-246: This paragraph is out of context, so I would suggest that the multiparametric FCM analysis (section 3.2) should appear before section 2.1 – as you explain things in lines 240-246 that is not shown yet.
Line 258: It is not clear what you intend with "the same distribution". This needs to be described. You need to describe in words what you see in Figure 2, before you state your interpretation of the data.
Figure 2: The histograms of PI staining should also be shown and also dot plots of SSC versus FSC should be shown. For starvation, you see a loss of SYTO 9 staining (which stains both live and dead bacteria). The reduced SYTO 9 staining can be due to cytoplasmic leakage of the stain, or increased activity of efflux pump which pumps SYTO 9 out of the cells (doi: 10.1039/d3an02112b). This has to be discussed. Bar graphs showing the geometric mean intensities at for the different parameters (SYTO 9, PI, FSC, SSC) in Figure 2 ought to be shown. Yellow is not a good color in graphs due to low visibility. Please use another color. Figure 3A belongs to Figure 2. Figures 3A and B belong to Figure 5.
Line 270-: As mentioned in the previous review, the geometric mean should be mentioned and not the maximum peak, since the geometric mean includes all events. Moreover, in some samples you have two peaks of SYTO 9 staining (e.g., 12 and 24 h).
Line 277: PI also interacts with RNA, and not always PI replaces SYTO 9.
Line 279- and line 294-: You continue describing without referring to "starvation conditions" or "hypoxia". Only in a later sentence hypoxia is mentioned. Since Fig. 3 contains hypoxia, this is quite confusing. I suggest reorganizing the result section to start with starvation and then hypoxia. In the current state there is a jump back and forth. (e.g., Lines 279-283 are about starvation, lines 294-300 are about hypoxia, and then lines 300- are about starvation). Please make subsections to better present the data.
Line 305: Then the authors speak about "anucleated cells" before showing their existence. A Figure should be mentioned here.
Reference should be added to the sentence of lines 304-305 showing the involvement of FtsZ and DnaA in cell division. Also, their specific function in cell division should be described.
The following can be deleted: ". Our research, which delves into this complexity,"
Lines 310: "These findings provide a deeper understanding of the intricate role of FtsZ and DnaA in bacterial growth and division and their implications in survival strategies under adverse conditions" belongs to discussion.
Figure 4 includes additional genes (dosR, hspX) not discussed in the text of Result section. Standard deviation lacks in B and D.
In figure legend 5, please write SYTO 9 (instead of SYTO). Also, in lines 453-459. And in lines 542-543.
Figure 6: The dimension of the size bar should be shown in the figure. The same for figure 7.
Line 426: Instead of "cell cycle" which is used for mammalian cells, state "cell division".
Line 427: Seems to be overinterpretation. What is the division time of these bacteria? Experiments have to be done to prove that "nutrients from dead cells trigger the growth of neighboring cells". You can simply collect the condition medium from dead cells and provide it to a hypoxic culture to see if there indeed is a revival.
The enlarged images in Figure 7 are not from the same site as the lower magnifications. The magnifications should be from the same site.
Line 429: FtsZ is required for septum formation, so how can reduced ftsZ expression lead to more septa? Could it be reduced autolysin activity?
The pleiotropic transformation needs to be described in better terms. A quantification of bacterial length/width/volume in a graph would provide a better information. The bacterial lengths in Table 2 do not fit with the images of Figure 8 – How was the length determined? You need to measure several hundreds of bacteria.
Figure 8: Please add to subfigure A an additional image.
Figure 9: Higher magnification of time 0 should also be shown. The size of scale bars should be stated in the figure.
Author Response
The manuscript has been improved. However, there are several issues that still need to be addressed before the manuscript can be accepted for publication.
Line 93: The definition of the two dormant phases NRP1 and NRP2 should be stated.
Reply: We now included the definition of NRP1 and NRP2 phases (lines 93-95)
For NRP1, the oxygen concentration in the culture was approximately 1% (hypoxia), while for NRP2, it was less than 0.06% (anaerobiosis).
Line 100: Please add space after the parenthesis.
Reply: We amended this
Lines 115-116 are a repetition of previous sentence. Please delete one of them.
Reply: We amended this sentence
Line 119-120: The final concentration of SYTO 9 and PI should be stated.
Reply: Final concentrations of both fluorophores have been added to lines 119-120
“ to this suspension, 6 µL of an equimolar solution of components A and B were added [component A (9.86µM SYTO 9 final concentration) and component B (59 µM Propidium Iodide final concentration)]”
Line 121: The composition of Kohn-Harris medium should be stated.
Reply: We now included the reference for the composition of the Kohn-Harris medium (Gonzalez-y-Merchand et al., 1998).
Sentence in line 167 is incomplete (lacks a verb).
Reply: We apologize for this, now we added “were obtained” as a verb
Line 174: I do not think you washed the RNA with Trizol reagent, it should be washed twice with 70% ethanol. Please check the accuracy of the text.
Reply: After RNA precipitation, we purified the isolated RNA with Trizol reagent three times to eliminate DNA remnants. Now we changed the term washed to purified for better understanding.
Section 2.9: The sequence of all primers should be presented.
Reply: We added this information to this section, lines 186-190.
Line 210: The morphological changes are not shown in Figure 1. Please show them and describe which morphological changes have occurred.
Reply: We apologize for this, and we rephrased this paragraph as:
"Our study reveals that M. smegmatis mc2 155 survived and persisted during 120 h of starvation or hypoxic conditions (Figure 1); during this time, several viability and morphological changes occurred: a membrane compromise of the M. smegmatis was identified by FCM experiments (Figures 2 and 3), cell length reduction and the appearance of pleomorphic and rough cells (Table 1 and 2)".
Figure 1 lacks the A and B labeling in the figure itself. I would also suggest adding "starvation" and "hypoxia" above the respective subfigures.
Reply: We apologize for this; we have amended this figure, and the legends you kindly suggested are included.
Figure 1B, green graph at around 36 h, the stdev is only in one direction (minus), the plus stdev is lacking.
Reply: Both directions of the SD are present at 36 h (the plus direction is small). Now, we reduced the width of the green line for better observation, as shown in Figure 3B.
Lines 224- : It is important to refer in the text to Figure 1A.
Reply: Figure 1A was referred in the line 213
Line 238: You have to describe in words the difference observed, and the statistical significance has to be added. Statistical significance should also be indicated in Figure 1A and B. How many times was Figure 1B repeated? This should be stated in the figure legend. Is this drop in SYTO 9 positive cells seen in all experiments of hypoxia?
Reply: As shown in Figure 1 A, the CFU/mL and v-bact determined by FCM were similar (with no significant difference). In Figure 2B, we identified a new population of bacteria do not present during starvation, which complicated the direct correlation in this figure. As indicated, these data represent the values carried out in triplicate, a practice that ensures the robustness and reliability of our experimental process. The consistent drop in SYTO 9 positive cells was a pattern observed across all hypoxia experiments.
Section in lines 240-246: This paragraph is out of context, so I would suggest that the multiparametric FCM analysis (section 3.2) should appear before section 2.1 – as you explain things in lines 240-246 that is not shown yet.
Reply: We believe that including the times at which NRP phases 1 and 2 are reached in the results section is essential, as this provides valuable context for analysing the other results. If we were to move this information to the methodology section, it would diminish the significance of these results.
Line 258: It is not clear what you intend with "the same distribution". This needs to be described. You need to describe in words what you see in Figure 2, before you state your interpretation of the data.
Reply: “the same distribution” was changed to populations that acquire the same behaviour based on staining with these fluorochromes. In the lines 265-266
Figure 2: The histograms of PI staining should also be shown and also dot plots of SSC versus FSC should be shown. For starvation, you see a loss of SYTO 9 staining (which stains both live and dead bacteria). The reduced SYTO 9 staining can be due to cytoplasmic leakage of the stain, or increased activity of efflux pump which pumps SYTO 9 out of the cells (doi: 10.1039/d3an02112b). This has to be discussed. Bar graphs showing the geometric mean intensities at for the different parameters (SYTO 9, PI, FSC, SSC) in Figure 2 ought to be shown. Yellow is not a good color in graphs due to low visibility. Please use another color. Figure 3A belongs to Figure 2. Figures 3A and B belong to Figure 5.
Reply: We would like to express our gratitude for your suggestion regarding efflux pumps. We have now incorporated this topic into the discussion (lines 369-371). Additionally, we have redistributed the panels of this figure to Figures 2 and 5, which is now Figure 3, and have improved the colour scheme.
Line 270-: As mentioned in the previous review, the geometric mean should be mentioned and not the maximum peak, since the geometric mean includes all events. Moreover, in some samples you have two peaks of SYTO 9 staining (e.g., 12 and 24 h).
Reply: We agree that the geometric mean is a more representative measure because it takes all events into account rather than just the maximum peak. Moving forward, we will prioritize the geometric mean in our analyses. Additionally, we will respond to the observation about the two peaks of SYTO 9 staining found in certain samples, specifically on lines 535 and 536.
Line 277: PI also interacts with RNA, and not always PI replaces SYTO 9.
Reply: We have changed this phrase to “by replacing it in the nuclei acid interaction”
Line 279- and line 294-: You continue describing without referring to "starvation conditions" or "hypoxia". Only in a later sentence hypoxia is mentioned. Since Fig. 3 contains hypoxia, this is quite confusing. I suggest reorganizing the result section to start with starvation and then hypoxia. In the current state there is a jump back and forth. (e.g., Lines 279-283 are about starvation, lines 294-300 are about hypoxia, and then lines 300- are about starvation). Please make subsections to better present the data.
Reply: We have restructured this paragraph and created Results section 3.3. Additionally, Figure 3 has been removed, and its data has been incorporated into Figures 2 and 3 (previously Figure 5).
Line 305: Then the authors speak about "anucleated cells" before showing their existence. A Figure should be mentioned here.
Reply: We have changed this mentioned as suggestion before we showed their existence
Reference should be added to the sentence of lines 304-305 showing the involvement of FtsZ and DnaA in cell division. Also, their specific function in cell division should be described.
Reply: Two references were included at the end of this sentence line 358. And the specific function of FtsZ and DnaA in cell division is mentioned in lines 354-355.
Manjot Kiran 1, Erin Maloney, Hava Lofton, Ashwini Chauhan, Rasmus Jensen, Renata Dziedzic, Murty Madiraju, Malini Rajagopalan. Mycobacterium tuberculosis ftsZ expression and minimal promoter activity. Tuberculosis (Edinb). 2009 Dec;89 Suppl 1(Suppl 1):S60-4. doi: 10.1016/S1472-9792(09)70014-9.
Anna Zawilak 1, Agnieszka Kois, Grazyna Konopa, Aleksandra Smulczyk-Krawczyszyn, Jolanta Zakrzewska-Czerwińska. Mycobacterium tuberculosis DnaA initiator protein: purification and DNA-binding requirements. Biochem J. 2004 Aug 15;382(Pt 1):247-52. doi: 10.1042/BJ20040338.
The following can be deleted: ". Our research, which delves into this complexity,"
Reply: We eliminated this sentence
Lines 310: "These findings provide a deeper understanding of the intricate role of FtsZ and DnaA in bacterial growth and division and their implications in survival strategies under adverse conditions" belongs to discussion.
Reply: It has been removed from the results section, and we stated these findings in the discussion section (lines 517-521).
Figure 4 includes additional genes (dosR, hspX) not discussed in the text of Result section. Standard deviation lacks in B and D.
Reply: We have now included information about the gene expression of dosR and hspX in Results section 3.3. The standard deviation is not shown, as these images display relative gene quantification as a ratio comparing stress conditions to the exponential phase.
Likewise, under hypoxic conditions at 12 h, it was observed that the dosR and hspX genes were over-expressed during hypoxia, as time passed, the expression of these genes decreased (Figure 4 C and D), observing that dosR and hspX genes were expressed 1 x106 fold. These genes are expressed when the mycobacteria under state of hypoxia, dosR hypoxia regulator, ensures the survival of the mycobacteria during hypoxia-induced in vitro latency; while the hspX gene is induced in low oxygen tension, the HspX protein also helps the mycobacteria to survive during non-replicative persistence phases.
In figure legend 5, please write SYTO 9 (instead of SYTO). Also, in lines 453-459. And in lines 542-543.
Reply: We have changed SYTO by SYTO9 in all cases
Figure 6: The dimension of the size bar should be shown in the figure. The same for figure 7.
Reply: Now we included this information in the figure
Line 426: Instead of "cell cycle" which is used for mammalian cells, state "cell division".
Reply: We amended this
Line 427: Seems to be overinterpretation. What is the division time of these bacteria? Experiments have to be done to prove that "nutrients from dead cells trigger the growth of neighboring cells". You can simply collect the condition medium from dead cells and provide it to a hypoxic culture to see if there indeed is a revival.
Reply: The division time according to the growth curve performed in our laboratory was 2.33 h (data not shown). The reviewer's proposal to see if "the nutrients from dead cells trigger the growth of neighboring cells" is interesting, but it is not part of the objective and hypothesis of the work.
The enlarged images in Figure 7 are not from the same site as the lower magnifications. The magnifications should be from the same site.
Reply: We apologize for this, but we don’t have magnifications from the same site.
Line 429: FtsZ is required for septum formation, so how can reduced ftsZ expression lead to more septa? Could it be reduced autolysin activity?
Reply: Downregulation of FtsZ leads to long undivided cells containing the septum into the midcell Z-rings. However, the constriction force, which is the mechanical force exerted by the Z-rings to divide cells, cannot be realized due to the low amount of the FtsZ protein.
The pleiotropic transformation needs to be described in better terms. A quantification of bacterial length/width/volume in a graph would provide a better information. The bacterial lengths in Table 2 do not fit with the images of Figure 8 – How was the length determined? You need to measure several hundreds of bacteria.
Reply: The lengths of the cells shown in Fig 8 correspond only to those that maintain normal road morphology. A minimum of 50 cells with normal morphology were measured, as indicated below the table. It's important to note that these measurements should ideally be obtained from multiple bacteria, but this is not feasible at the moment.
Figure 8: Please add to subfigure A an additional image.
Reply: We amended this
Figure 9: Higher magnification of time 0 should also be shown. The size of scale bars should be stated in the figure.
Reply: We now included an additional figure of time 0 h and the size of the scale bars.
Reviewer 2 Report
Comments and Suggestions for Authors
The authors have satisfactorily responded to all my questions and made the necessary changes to the manuscript
Author Response

(The authors gave the same response as above.)
